# Multiscale flood risk assessment under climate change: the case of the Miño River in the city of Ourense, Spain

Diego Fernández-Nóvoa[1,2], Orlando García-Feal[1,3], José González-Cao[1], Maite deCastro[1], Moncho Gómez-Gesteira[1]

[1]Centro de Investigación Mariña (CIM), Universidade de Vigo, Environmental Physics Laboratory (EPhysLab), Campus da Auga, 32004 Ourense, Spain
[2]Instituto Dom Luiz (IDL), Faculdade de Ciências da Universidade de Lisboa, 1749-016 Lisbon, Portugal
[3]Water and Environmental Engineering Group, Department of Civil Engineering, Universidade da Coruña, A Coruña, Spain

*Correspondence to*: Diego Fernández-Nóvoa (diefernandez@uvigo.es)

**Abstract.** River floods, which are one of the most dangerous natural hazards worldwide, have increased in intensity and frequency in recent decades as a result of climate change, and the future scenario is expected to be even worse. Therefore, their knowledge, predictability, and mitigation represent a key challenge for the scientific community in the coming decades, especially in those local areas that are most vulnerable to these extreme events. In this sense, a multiscale analysis is essential to obtain detailed maps of the future evolution of floods. In the multiscale analysis, the historical and future precipitation data from the CORDEX project are used as input in a hydrological model (HEC-HMS) which, in turn, feeds a 2D hydraulic model (Iber+). This integration allows knowing the projected future changes in the flow pattern of the river, as well as analyzing the impact of floods in vulnerable areas through the flood hazard maps obtained with hydraulic simulations. The multiscale analysis is applied to the case of the Miño-Sil basin (NW Spain), specifically to the city of Ourense. The results show a delay in the flood season and an increase in the frequency and intensity of extreme river flows in the Miño-Sil basin, which will cause more situations of flooding in many areas frequented by pedestrians and in important infrastructures of the city of Ourense. In addition, an increase in water depths associated with future floods was also detected, confirming the trend for future floods to be not only more frequent but also more intense. Detailed maps of the future evolution of floods also provide key information to decision-makers to take effective measures in advance in those areas most vulnerable to flooding in the coming decades. Although the methodology presented is applied to a particular area, its strength lies in the fact that its implementation in other basins and cities is simple, also taking into account that all the models used are freely accessible.

## 1 Introduction

A large amount of studies focused on the analysis and understanding of river floods have been developed in the last years (Benito et al., 2015; Portugués-Mollá et al., 2016; Zhu et al., 2017; Munoz et al., 2018; Areu-Rangel et al., 2019; Stoleriu et al., 2020; Mel et al., 2020; Di Sante et al., 2021). This prolific field of study is related to the fact that floods are considered as one of the most dangerous natural hazards occurring in the planet (Noji, 2000; Paprotny et al., 2018). In fact, several reports stated the potential of flood events to cause important damages in critical human infrastructures even putting at risk the integrity of people (Alderman et al., 2012; Fekete et al., 2017). Good indicators of the danger associated with floods are the estimated 100,000 deaths and the 1.4 billion people affected in the last decade of the 20th century (Joknman, 2005) or the billions of dollars in losses due to the damage in infrastructures (Wallemacq et al., 2018). Despite the incredible damage associated with flooding in the past, the future is expected to be worse. In fact, one of the most disturbing implications associated with climate change is the general increase in the river flood hazard in the next decades (Dankers and Feyen, 2008; Arnell and Gosling, 2016; Alfieri et al., 2017; Jongman, 2018). However, these authors also indicate that there is considerable uncertainty regarding the magnitude of future flood risks on regional and local scales. In this sense, climate reports highlight that climate extremes, such as floods, occur first at the local level, affecting the local population and local places (Cutter et al., 2012). This also reveals the need to know and understand the future implications of these events on a regional and local scale, especially in vulnerable areas. In fact, mitigating the impact of floods represents one of the most important challenges facing the scientific community worldwide in the coming decades (IPCC, 2012). The Iberian Peninsula is not foreign to disasters provoked by river floods, especially over its western area (Zêzere at al., 2014; Pereira et al., 2016). This risk was demonstrated in several studies that analyze the dramatic consequences suffered by overflowing of the Iberian Rivers debouching into the Atlantic Ocean (Benito et al., 1996; 2003; Ortega et al., 2009; Trigo et al., 2014; 2016; Pereira et al., 2016; Rebelo et al., 2018; González-Cao et al., 2021). The special synoptic conditions of the Iberian Peninsula, where the storm-tracks in the Northern Hemisphere can transport heat and moisture, are able to promote extreme weather conditions over this area (Peixoto and Oort, 1992; Trigo, 2006). In particular, although the hydrological cycle regimes in the Iberian Peninsula are conditioned by several atmospheric variability modes (deCastro et al., 2006), the timing and position of winter storms are highly dependent on the North Atlantic Oscillation (NAO) phase, whose positive mode favors that the western Iberian Peninsula can be subjected to continuous intense large-scale precipitation during winter months, the main cause of the floods developed in the major rivers of this area (Lavers et al., 2011; Trigo et al., 2014). In this sense, some studies analyzing the future evolution of precipitation over Spain have shown an increase in extreme precipitation events (Lorenzo and Alvarez, 2020; Des et al., 2021), which reinforces the need to analyze and understand how this increase will condition flooding of rivers in vulnerable areas.

For all the above mentioned, the main aim of this study is focused on the analysis of the future evolution of risk river flows in the regional domain delimited by the international Miño-Sil basin (NW Spain), and specifically, on the associated floods caused in the city of Ourense. Ourense is the local domain where flooding of the Miño river can cause significant damages (Fernández-Nóvoa et al., 2020). For that, the integration of hydrological-hydraulic models, together with specific information

on flood thresholds in the area under scope, will allow a detailed analysis of those particular events capable of causing a significant impact in the study area. In this sense, most of the previous studies dealing with flood projections analyzed the expected changes in extreme flows and floods associated with different probabilities or return periods (Te Linde et al., 2011; Huang et al., 2013; Arnell and Gosling, 2016; Alfieri et al., 2017; Padulano et al., 2021). Here, the particular flows that effectively cause damage in Ourense city, not necessarily associated with characteristic return periods, as well as the specific areas that will be most affected by the associated floods, will be analyzed. This approach allows the analysis to be adapted to the particular characteristics of the area under scope, thus contributing to addressing one of the most important challenges facing the scientific community for the coming decades: assessing the implications of climate change on extreme events at the local level. To achieve these objectives, the river flows in the Minho-Sil basin will be simulated for an entire historical period of 30 years and also for an entire future period of 30 years using data provided by climate models, analyzing not only the changes in the probability of risk flow situations but also the evolution of flood risk in particular areas of the city. The results will provide detailed information to decision-makers to help them take accurate and effective measures to mitigate the damages associated with future extreme events. In addition, it is important to highlight that although the methodology developed is focused on the Miño-Sil river basin, it can be generalized to any basin and location, showing its potential in future applications.

## 2 Study area, Data and Methods

This section describes the area under study, as well as the data used and the methodology carried out to analyze the future evolution of extreme river flows and flood events in the city of Ourense, inside the Miño-Sil basin.

### 2.1 Area of study

The area under scope is located in northwestern Iberian Peninsula (Figure 1a). It corresponds to the International Miño-Sil basin upstream Ourense city, encompassing more than 70% of the entire catchment, occupying near to 13,000 km$^2$ (Figure 1b). The basin under analysis ranges in altitude approximately from 110 to 2100 m.a.s.l., with an average slope of around 9.57º (Table 1). The basin presents an important variability of land uses, although it is mainly characterized by moors and heathland (25 %), broad-leaved forest (23 %) and complex cultivation patterns (17 %), according to the criteria provided by CORINE land cover data (CLC, 2018) (Figure 2). Special attention is focused on the Ourense city, where Miño river floods can cause important damage (Figure 1c). The Miño river, which has an approximate length of 134 km to Ourense, presents a pluvial regime, with an annual hydrologic cycle characterized by minimum river flows during summer months and maximum flows at the end of autumn and winter, when the extreme flood events can occur (Fernández-Nóvoa et al., 2017; 2020). This river flow cycle is promoted by the precipitation pattern of the study area, whose seasonal variability depends on the position of the Azores High and the Iceland Low, which determine the occurrence of most of the rainy events during the late autumn and

winter (deCastro et al., 2006). When the Miño river flow surpasses a certain level at Ourense city a risk is generated in areas frequented by pedestrians and in some important infrastructures (Fernández-Nóvoa et al., 2020).

## 2.2 Precipitation data

Historical (1990-2019) and Future (2070-2099) daily precipitation data for the area under scope were retrieved from the Regional Climate Models (RCMs) simulations carried out in the framework of the CORDEX project (http://www.euro-cordex.net/). EURO-CORDEX initiative offers simulations over the European continent considering global climate data from the Coupled Model Intercomparison Project Phase 5 (CMIP5) up to the year 2100, with more than 30 RCMs corresponding to the RCP8.5 greenhouse gas emission scenario. In this sense, some studies have detected that greenhouse gas emissions in recent years have tracked most closely with those projected under RCP 8.5, so this emission scenario seems to be highly realistic and a useful tool to assess future climate risks (Schwalm et al., 2020). EURO-CORDEX models provide daily data with sufficient spatial resolution ($0.11° \times 0.11°$) to adequately address the hydrological procedures of the area of scope (Garijo and Mediero, 2018; Lorenzo and Alvarez, 2020; Des et al., 2021).

Daily measured precipitation data were obtained from the pluviometers managed by MeteoGalicia (the Regional Meteorological Agency of Galicia, www.meteogalicia.gal) to validate the CORDEX models for the study area. The measured precipitation was downloaded for the period 2008-2020, which is available for most of the pluviometers located in the area under scope.

## 2.3 Hydrological and Hydraulic Models

The transformation of precipitation into the corresponding river flow was carried out using the semi-distributed model HEC-HMS (Feldman, 2000; Scharffenberg, 2018; U.S. Army Corps of Engineers, 2018). This hydrological model was selected for being one of the most robust and widely adopted for hydrological procedures worldwide. In addition, this model has shown accurate results for the basin under scope as well as other nearby basins (Cea and Fraga, 2018; González-Cao et al., 2019; Fraga et al., 2020; Fernández-Nóvoa et al., 2020). In particular, Fernández-Nóvoa et al. (2020) shown an effective hydrological procedure for the area under scope using HEC-HMS and considering: i) the Soil Conservation Service (SCS) curve number for the rainfall infiltration calculations; ii) the SCS unit hydrograph to convert the rainfall excess in surface runoff; iii) the linear reservoir methodology to simulate the corresponding baseflow; iv) the Muskingum-Cunge method to deal with runoff propagation along the main channels. HEC-HMS is freely available on its official website (https://www.hec.usace.army.mil/software/hec-hms/).

The hydraulic model Iber+ was used to evaluate floods in Ourense city. Iber+ is a numerical tool that solves the 2D depth-averaged shallow water equations applying the finite volume method (García-Feal et al., 2018), and it is freely available on its official website (http://iberaula.es). In particular, the equations resolved by Iber+ model can be written as follows:

$$\frac{\partial h}{\partial t} + \frac{\partial hU_x}{\partial x} + \frac{\partial hU_y}{\partial y} = 0 \tag{1}$$

$$\frac{\partial hU_x}{\partial t} + \frac{\partial}{\partial x}\left(hU_x^2 + g\frac{h^2}{2}\right) + \frac{\partial}{\partial y}\left(hU_xU_y\right) = -gh\frac{\partial Z_b}{\partial x} - \frac{\tau_{b,x}}{\rho} + \frac{\partial}{\partial x}\left(v_t h\frac{\partial U_x}{\partial x}\right) + \frac{\partial}{\partial y}\left(v_t h\frac{\partial U_x}{\partial y}\right) \tag{2}$$

$$\frac{\partial hU_y}{\partial t} + \frac{\partial}{\partial y}\left(hU_y^2 + g\frac{h^2}{2}\right) + \frac{\partial}{\partial x}\left(hU_xU_y\right) = -gh\frac{\partial Z_b}{\partial y} - \frac{\tau_{b,y}}{\rho} + \frac{\partial}{\partial x}\left(v_t h\frac{\partial U_y}{\partial x}\right) + \frac{\partial}{\partial y}\left(v_t h\frac{\partial U_y}{\partial y}\right) \tag{3}$$

where $h$ is the water depth, $U_x$ and $U_y$ represent the averaged horizontal velocities, $g$ is referred to the acceleration of the gravity, $\rho$ is the density of the water, $v_t$ is the turbulent viscosity, $Z_b$ is the bed elevation, and $\tau_b$ represents the bed friction. The bed friction is computed with the Manning formulation as:

$$\tau_{b,x} = \rho g h \frac{n^2 U_x |U|^2}{h^{4/3}} \qquad \tau_{b,y} = \rho g h \frac{n^2 U_y |U|^2}{h^{4/3}} \tag{4}$$

Iber+ is an implementation in C++ and CUDA of the Iber model (Bladé and Cea, 2014). This new and optimized code achieves a two-order of magnitude speed-up by using graphical processing unit (GPU) and high-performance computing (HPC) techniques. These improvements allow to overcome the time constrained limitations of this type of climatological studies which require a large number of simulations. Iber+ has shown to provide accurate results in several studies conducted in the area under the scope and in nearby areas (González-Cao et al., 2019; Fraga et al., 2020; Fernández-Nóvoa et al., 2020).

The good performance of both models to resolve hydrological and hydraulic procedures for the study area allowed their integration for the development of an Early Warning System over the Miño-Sil basin with a good capability to reproduce real events (Fernández-Nóvoa et al., 2020). Therefore, attending to the results obtained in this previous study, where both models were calibrated and successfully validated for the study area, the same configuration of both models detailed in Fernández-Nóvoa et al. (2020) was maintained to develop the present study. The catchment schematization followed is presented in Figure 3.

## 2.4 Methodology

The capability of each of the more than 30 RCMs models to represent precipitation over the area under the scope was tested comparing RCMs precipitation data and field data from pluviometers managed by MeteoGalicia for the common period 2008-2020. Although there are several valid approaches to validating CORDEX models (Perkins et al., 2007; Peres et al., 2020; Des et al., 2021) the present analysis is focused on testing both the entire distribution of precipitation data, the so called Perkins' test (Perkins et al., 2007) and also the extreme precipitation values (those above the 99 percentile, $P^{99}$ test). One of the main advantages of the Perkins' test, based on probability density functions and developed on a daily scale, is that the good fit between the measured and simulated data proves the capability of the model to detect extreme values that can be unusual in the historical period but more common in the future due to the implications of climate change (Perkins et al., 2007). This

allows selecting the most appropriate models according to the scope of this study, since the extreme precipitation events are

145 usually behind the most important river floods. In addition, the complementary analysis focused on evaluating the deviation of the models in relation to the extreme precipitation values reinforces the validation process. In this sense, only those models surpassing 90% in the Perkins' test and with a deviation less than 25 % in the extreme values were considered. In addition, in order to reinforce the detection of those models especially suitable for representing extreme events in the area under scope, the comparison was carried out over the wet season (November-March) when the flood events take place (Fernández-Nóvoa et

al., 2020).

Once the models that provide a good characterization of the precipitation for the study area have been selected, the precipitation provided by each one is used as an input in the hydrological model to simulate the river flow in the entire Miño-Sil basin. Thus, a hydrological simulation was carried out for each valid CORDEX model considering both historical (1990-2019) and future (2070-2099) periods. Although precipitation data from CORDEX is on a daily scale, it was added to the hydrological

model on an hourly scale (dividing daily data by 24) in order to obtain river flows at this scale. Therefore, the historical and future flows of the river were obtained on an hourly scale. It is important to point out that a day in a flood alert situation is considered when the hourly river flow at the city of Ourense surpasses the 2000 $m^3s^{-1}$ at some moment. According to previous research (Fernández-Nóvoa et al., 2020), this threshold marks the limits above which Miño river can cause dangerous flooding in areas frequented by pedestrians and serious damage to infrastructures. Changes in alert flood situations between historical

and future periods were evaluated for each valid model. This analysis was carried out considering the total number of days under flood risk situation detected throughout the historical period (1990-2019) and the changes expected in the future period (2070-2099). The possible variation in the seasonality of the floods has also been evaluated. In addition, the river flow series obtained for the entire historical and future periods considered also allow the analysis of the expected changes in the general variability of river flow. Finally, taking advantage of the Iber+ model, an in-depth analysis of the dynamic evolution of future

floods in different areas of the city has also been developed, which adds detailed knowledge of the future impact of floods in critical areas of the city. Some of the results will also be shown by averaging the individual results obtained after forcing the hydrological-hydraulic models with the data provided by each valid CORDEX model (multi-model), which minimizes the biases and uncertainties of each model (Pierce et al., 2009; Jacob et al., 2014).

The general architecture of the methodology developed is schematized in Figure 4. The developed procedure takes between 2-

170 3 weeks to execute each model despite the improvements in the numerical procedures mentioned above (with the simulations executed on a computer with an AMD Ryzen 7 2700X processor, 32 GB of RAM and a Nvidia RTX 3080 ti GPU). However, this execution supposes an important added value in relation to this type of study. Most of the previous works that analyze changes in flood risk as a consequence of climate change are limited to the hydrological analysis, showing in a general way the expected future changes in high flows of the river generally associated with different return periods. However, it is

175 necessary to assess the particular impacts of future flooding in vulnerable areas. In this work, the methodology presented allows a more detailed analysis of those flows that effectively cause flooding in a particular area, evaluating the expected

changes as a result of climate change, as well as the risk associated with future flooding in especially important and vulnerable areas of the cities.

## 3 Results and Discussion

### 3.1 RCMs precipitation validation

Precipitation data of the available RCMs models were evaluated in order to select those models whose precipitation represents accurately the characteristics of the area under the scope. Table 2 shows the performance of models when Perkins and P[99] tests were applied. Only those models surpassing both conditions were taken into account (for sake of clarity a number was assigned to each model). Six models meet both conditions and were selected to develop the study (Models 4, 6, 13, 14, 26 and 29. See Table 2). The precipitation associated with each of these models was used to feed the hydrological model in order to obtain river flows for historical and future periods.

### 3.2 Future variability in river flow pattern

The future evolution in the river flow distribution was characterized for each valid model by the percentage of change in the frequency of the river flow ranges with respect to the historical period (Figures 5 and 6). For this, the river flow data was distributed in ranges of 200 $m^3s^{-1}$, considering in the last range those situations that surpassed the alert threshold. The number of cases within each range was counted both for the historical and future periods and the percentage of change for each range was calculated according to the following equation:

$$Increment\ (\%)_r = \frac{F_r - H_r}{H_r} *100 \qquad (5)$$

where $r$ is the respective range, $F$ is the number of cases in the future period, and $H$ is the number of cases in the historical period. Positive (Negative) values indicate an increase (decrease) in the number of cases in each flow range in the future. Multimodel mean shows that the frequency of the low ($< 400$ $m^3s^{-1}$) and high ($> 1600$ $m^3s^{-1}$) river flows will be increased in the future, whereas the intermediate flows will tend to decrease (Figure 6). In particular, models 4 and 6 show a future increase in the frequency of river flows above 1200 $m^3s^{-1}$ while in models 14 and 29 this increase is observed from 1800 $m^3s^{-1}$, and in models 13 and 26 an increase is only detected for the most extreme range, those values above the flood alert limit of 2000 $m^3s^{-1}$ (Figure 5). With respect to lower flows, the models indicate an increase in the frequency of flows below 400 $m^3s^{-1}$, except in models 4 and 6 where the only river flows that will increase their frequency are those below to 200 $m^3s^{-1}$ (Figure 5). In any case, this supposes an intensification of the hydrological cycle with a possible increase in both, the drought and flood situations, an effect that has been also detected in other river basins around the world (Dery et al., 2009; Gloor et al., 2013; Davis et al., 2015; Gouveia et al., 2019). This future distribution of the river flow regime would suppose, for the area of study, more situations of water deficit during the dry season and a greater risk of flood situations during the wet season.

The river discharge, both for the historical and future, was also analyzed for different percentiles of flow (Table 3). A general decrease in the river discharge values associated to the different percentiles is observed in all models for the future, except for the highest percentiles ($> 95^{th}$). At this point, it is important to keep in mind that the $50^{th}$ percentile represents river flows of the order of 200 m³s⁻¹, and all the models presented a future increase in the frequency of flows below this value, as can be seen in Figures 5 and 6. This explains the general decrease of river discharge in these percentiles, and supports the idea that low flows will be more frequent in the future, which could condition the water availability. On average, the models indicate a decrease of about 10% in the $50^{th}$ percentile. The opposite situation occurs when the highest river flows are analyzed. In fact, only the highest percentiles present an increase in river discharge in the future. Thus, the flow rate at the $99.9^{th}$ percentile increases in all models. This means that the extreme flows of the river, those capable of causing floods, will be intensified in the future, which corroborates the results presented in Figures 5 and 6. On average, the models indicate an increase in the flow of about 15% at this extreme percentile. In addition, model 14 shows an increase in flow from the $99.5^{th}$ percentile and even some models present an increase from the $99^{th}$ (Models 4, 6, and 13, Table 3). In any case, taking into account that floods in Ourense begin to be critical from 2000 m³s⁻¹, all the models indicate an increase of these extreme flows, corroborating the increase in the intensity and frequency of critical flows in Ourense in the future.

## 3.3 Future evolution of flood alerts

The evolution of flood alerts in Ourense was evaluated considering the number of days in a flood alert situation (river flow surpassing 2000 m³s⁻¹ at some moment of the day) for the historical and future periods (Table 4). All models indicate an increase in the number of days under a flood alert situation in the future. Particularly, models 6 and 13 show results especially remarkable, as they indicate an increase above 60 % in the number of days under flood alert for the future. At the opposite extreme, models 4 and 29 indicate a more moderate increment of about 10 %. On average, the models indicate 22 ± 7 days in flood alert situation in the historical period and that in the future, the alert situations will be increased to 30 ± 11 days. This means an average increase of more than 35% in the number of days under flood risk situations in the future as a consequence of changes in precipitation patterns caused by climate change. In addition, the intensity of flood extreme events seems to be also increased. The mean flood discharge value, calculated averaging only those flows surpassing the alert threshold, was evaluated for the historical and future periods. On average, models have shown a mean flood discharge value of 2536 ± 198 m³s⁻¹ for the historical period incremented by a 7 % to reach 2713 ± 234 m³s⁻¹ in the future. The same procedure was carried out considering the maximum values reached in each model. On average the maximum peak value for historical period was 3743 ± 864 m³s⁻¹, increasing to 4765 ± 1233 m³s⁻¹ in the future period, which supposes an increment of approximately 27 %. These results imply an increase in both the frequency and intensity of floods in the city of Ourense which is consistent with previous studies of a more global nature. Alfieri et al. (2017) stated an increase in the risk of flooding in most of the world as a consequence of global warming. Furthermore, Kundzewicz et al. (2010) concluded that larger floods will be more common in the future over a large part of the European continent. However, as pointed in the introduction, it is necessary the

development of regional and local studies to confirm the global and general trends in regions and cities particularly vulnerable to flood risks.

## 3.4 Future flood seasonality

The impact of climate change on the seasonality of the river floods in Ourense was evaluated through the monthly analysis of the multi-model frequency of the floods for the historical and future periods and by the difference between both periods (Figure 7). 75% of flood events occur in December (51%) and January (24%) months for the historical period (Figure 7a). On the contrary, a significant decrease in the occurrence of floods is detected in December in the future (Figures 7b and c), while January and February will present a significant increase in the frequency of floods. This means that the flood season will be delayed in the future. January will be the month in which more floods will occur (40%) followed by February with 25%, and December with 24%. Furthermore, floods are absent between April and October in both historical and future periods. Similar results were obtained in previous works analyzing floods in the Iberian Peninsula (Blöschl et al., 2017). Likewise, Garijo and Mediero (2018) detected a future delay in the flooding season in some sub-basins of Ebro river. This means that climate change will not only increase the frequency and intensity of floods in the Miño-Sil basin, but will also produce changes in the flood season, which will be delayed in the future.

## 3.5 Future flood impacts in Ourense city

The delimitation of those areas of the cities most vulnerable to floods under the future scenario is essential when taking adequate and precise measures to mitigate the impacts of floods at the local level. In the particular case of the city of Ourense, every day that surpasses the pre-alert threshold (1253 $m^3$s, following Fernández-Nóvoa et al., 2020) was simulated with Iber+ for each precipitation valid model both for historical and future periods. This implies performing more than 1000 hydraulic simulations of 24-hour events in the area of the city of Ourense. In order to execute this large number of simulations in a reasonable time, the GPU parallelization of Iber+ was fundamental, as commented above. This flow threshold was selected for hydraulic simulations because some areas outside the riverbed can be reached by water when this limit is surpassed, thus providing a better representation of all flooded areas, which facilitates and improves the visual analysis of the hydraulic results. In this sense, the multi-model mean of the days under hazard by flood is calculated for each pixel of the domain (excluding the riverbed) for the historical period (Figure 8). The criteria for defining flood hazard situations was established following Cox et al. (2010). As expected, the areas closest to the riverbed present a higher frequency of flood hazards. However, the areas farthest from the riverbed, where there are some important infrastructures such as roads, parks, or thermal baths, are also threatened by floods, although less frequently. Changes in hazard flood situations between historical and future periods were evaluated in Ourense (Figure 9). This analysis was performed by subtracting the multi-model mean of the number of days in a hazard flood situation obtained in the historical period from those expected in the future. In the future, there will be a general increase in the hazard (red colors) in all areas affected by floods, which corroborates the results presented above (Figure 9a).

The area of most concern is the one marked in dark red, which shows a significant increase in the threat of flooding in highly frequented leisure areas. This area will be flooded due to extreme events. Although the areas closest to and farthest from the riverbed present a moderate increase (light red color) in the threat of flooding, in the latter its importance lies in the fact that the flooding is due to highly extreme events, corroborating the increase of these extreme flood situations. The light red color of the farthest area from the riverbed indicates a moderate increase in absolute days but a significant proportional increase in frequency and also in severity, since the area affected by floods is increased compared to historical one. These results are in agreement with the previous analysis of the flow patterns of the river.

Two of the most sensitive places in the city are analyzed in greater detail, showing the potential of the hydraulic model to delimit the vulnerable areas to flooding (1 and 2 in Figure 9a). Thus, an important increase in future hazard situations was detected in Chavasqueira thermal baths, one of the most visited places of the city (Figure 9b). This implies that the thermal baths will be closed during more days in the future as a result of flooding. This, along with an increase in costs associated with repairs due to flood damage, will have a negative socio-economic impact on the city. A similar analysis was made for the Oira park, one of the most important places of recreation in the city (Figure 9c). This area presents the greatest flood risk increase of the city in the future.

The multi-model mean of the maximum water depth reached during the floods in significant areas of Ourense was also calculated both for the historical period and for the future (Figure 10). For this analysis, the maximum value reached by water depth in each area for each day under flood conditions was determined, and the mean of these maxima (Figure 10a) and the supremum (Figure 10b) were calculated for each model and then averaged over all models. A part of the Chavasqueira thermal baths (1) and Oira park (2), other important recreational areas such as Antena beach (3), Ribeira de Canedo (4) and Miño (5) parks were considered. Furthermore, N-120 road (6) was also included since it is one of the most important transport routes of the city. An increase both in the average of maxima and supremum of the water depths was obtained at all locations in the future, which means that floods will tend to worsen in Ourense since in addition to the increase in the number of events, these will be most severe. The increase in the maximum (supremum) water depth will be of around 8% (17%) for the first four areas. The Miño park and the N-120 road deserve a separate analysis since the increase in the water depth is proportionally more pronounced. Future flooding in these locations will have associated water depths more than twice that for the historical period. It should be noted that the values of the water depth reached in Miño park and the N-120 road are much lower than those of the rest of the areas considered because these locations are far from riverbed and water only reaches them under very extreme conditions of river flow. Therefore, small increments in water depth can imply a larger increase in terms of percentage. This important increase in the farthest areas is a good indicator that future floods will be more extreme. The changes predicted for the N-120 road are especially remarkable due to their importance in relation to transport, as commented above. In this area the maximum average depth of water detected for the historical period is about 15 cm, which can cause a moderate effect on traffic, but it is expected to increase in the future to a value of about 35 cm on average. This implies that future floods in this area could condition traffic, which would have serious consequences.

**4 Conclusions**

Several global studies have confirmed the increased risk of river flooding in recent decades as a consequence of changes in precipitation patterns promoted by climate change, and a worsening of floods is expected in the future. However, it is necessary
to transfer this general view to regional and local areas under potential risks, in order to provide detailed information that helps the respective authorities to take effective measures to reduce the impacts of floods. In this sense, a multiscale analysis like the one described in this study is essential to be able to obtain detailed flood hazard maps for the future with very high spatial resolution from RCMs precipitation. In this study, the changes in the risk of floods expected in the future as a consequence of climate change for the Miño-Sil basin, and in particular over the city of Ourense, have been analyzed using precipitation data
from the CORDEX project together with hydrological and hydraulic models. Results showed that changes in future precipitation patterns over the Miño-Sil river basin will lead to an increase in the frequency and intensity of extreme river flows, which can affect many localities near the river. In the particular case of Ourense, the city will suffer an increase of around 35 % in the frequency of flood events in the future. In addition, future flood events will also present higher peak flows, which indicate an increase in their intensity. The changes also affect the seasonality of floods that will be delayed in the future.
In any case, in the future, there will be more critical situations in many areas frequented by pedestrians and in important infrastructures, as shown by the flood hazard maps obtained with 2D hydraulic simulations. The detailed analysis carried out in the most important areas of the city affected by floods, also concludes that in addition to being affected by a greater number of hazard situations in the future, these flood events will be more intense since the associated water depths will also be greater. All these translates into a negative socio-economic impact on the city.
The knowledge of the risk of floods in the most vulnerable areas of the city in the future allows the anticipation necessary to take effective measures focused on mitigating the negative effects of these events. Finally, it is important to remark that, despite the methodology presented is applied here to a target area, Miño-Sil river basin and the city of Ourense, its implementation in other basins and cities is straightforward, taking also into account that all the models implemented are freely available, which shows the potential in future applications.
The present work has some caveats, namely that the river flow can be also dependent of other interactions. In this sense, the present study tries to represent and analyze the evolution of the natural river flow based on the expected changes in precipitation patterns, as well as the implications in terms of the future floods. However, river flow is also influenced by man-made interactions, such as changes in river margins, soil characteristics, water use or the hydroelectric management, among others. These factors, which are non-easy predictable, can also have influence on the flood evolution. In this sense, the present study
can suppose a basis to following works dealing with the analysis and inclusion, at some extent, of the influence of these other interactions in flood dynamics.

*Code and data availability:* Freely available data and software (Precipitation data from CORDEX and MeteoGalicia, and HEC-HMS and Iber+ models) were used for this work.

*Author contribution:* DFN: Conceptualization, Methodology, Formal analysis, Investigation, Writing – Original Draft. OGF: Conceptualization, Methodology, Investigation, Writing – Review & Editing. JGC: Methodology, Investigation, Writing –
Review & Editing. MdC: Conceptualization, Methodology, Writing – Review & Editing, Supervision. MGG: Conceptualization, Methodology, Writing – Review & Editing, Supervision.

*Competing interests:* The authors declare that they have no conflict interest.

**Acknowledgments**

The authors thank the WCRP's Working Group on Regional Climate, and the Working Group on Coupled Modeling, former coordinating body of CORDEX and responsible panel for CMIP5. We also thank the climate modeling groups for producing and making available their models' outputs that can be downloaded at http://www.cordex.org. The authors also thank MeteoGalicia for the precipitation data provided.
The aerial pictures used in this work are courtesy of the Spanish IGN (Instituto Geográfico Nacional) and part of the PNOA (Plan Nacional de Ortografía Aérea) program.

This work was partially funded by the European Regional Development Fund under the INTERREG-POCTEP project RISC_ML (Code: 0034_RISC_ML_6_E). This work was also partially financed by Xunta de Galicia, Consellería de Cultura, Educación e Universidade, under Project ED431C 2021/44 "Programa de Consolidación e Estructuración de Unidades de
Investigación Competitivas".

DFN was supported by Xunta de Galicia through a post-doctoral grant (ED481B-2021-108).

OGF was funded by Spanish "Ministerio de Universidades" and European Union – NextGenerationEU through the "Margarita Salas" post-doctoral grant.

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

**Figure Captions**

**Figure 1.** (a) Location of the Miño-Sil basin upstream of Ourense city. (b) Map of elevations of the catchment under the scope. The channel that flows from north to south represents the Miño river and the channel that flows from east to west its main tributary, the Sil river. (c) Area of study in Ourense city (PNOA courtesy of © Instituto Geográfico Nacional).

**Figure 2:** General land uses defined by Corine Land Cover and their distribution for the area under scope.

**Figure 3:** Map with the hydrological characterization of the basin under scope attending to its topographical features. *Subbasin* is referred to the divisions of the basin under scope, *Junction* is referred to the points in the flow network where multiple inflows combine, *Downstream* is referred to the point in the main river channel where each subbasin flows, and *Reach* represents the main river channels.

**Figure 4.** Flow chart of the developed methodology.

**Figure 5.** Percentage of change in the frequency of river flow ranges in the future with respect to the historical period for each valid model. The river flow data was distributed in ranges of 200 $m^3s^{-1}$. Each value of the river discharge indicates the mean of the corresponding range (100 $m^3s^{-1}$ corresponds to the range of 0-200 $m^3s^{-1}$). In the last range (2100 $m^3s^{-1}$), the river flows surpassing the alert threshold (2000 $m^3s^{-1}$) are included. Positive (negative) values indicate increase (decrease) in the future.

**Figure 6.** Percentage change in the frequency of river flow ranges in the future with respect to the historical period considering the average of valid models (multi-model). The river flow data was distributed in ranges of 200 $m^3s^{-1}$. Each value of the river discharge indicates the mean of the corresponding range (100 $m^3s^{-1}$ corresponds to the range of 0-200 $m^3s^{-1}$). In the last range (2100 $m^3s^{-1}$), the river flows surpassing the alert threshold (2000 $m^3s^{-1}$) are included. Positive (negative) values indicate increase (decrease) in the future. Black bars represent the multi-model deviation.

**Figure 7.** Multi-model mean of the monthly flood frequencies for the (a) historical and (b) future periods. Frequencies were calculated dividing the flood alerts detected in each month by the total number of alerts. Thus, the sum of the frequencies of all months is equal to 1. (c) Differences in flood frequency between historical and future periods. Positive (negative) values indicate an increasing (decreasing) of flood frequency in the future. Grey shadow areas represent multi-model deviation.

**Figure 8.** Multi-model mean of the number of days under hazard by floods for each pixel of the domain in the historical period. Only the area outside the riverbed was considered for the flood risk analysis. (Aerial picture courtesy of PNOA © Instituto Geográfico Nacional).

**Figure 9.** (a) Future variability in the number of days under hazard by floods in Ourense city. The increase (decrease) in the number of cases is obtained subtracting for each pixel of the domain the number of days under flood hazard situation in the historical period from the hazard cases in the future period. Thus, red (blue) colors represent the increase (decrease) in the flood hazard considering the average of all models. Only the area outside the riverbed was taken into account for the flood risk analysis. In lower panels two important areas of the city are shown in detail, (b) Chavasqueira thermal baths and (c) Oira park. (Aerial pictures courtesy of PNOA © Instituto Geográfico Nacional).

**Figure 10.** (a) Mean of maxima and (b) supremum water depths in important areas of Ourense for the historical (blue) and future (red) periods considering the average of all models. Areas under analysis are named from 1 to 6: Chavasqueira thermal baths, Oira Park, Antena fluvial beach, Ribeira de Canedo park, Miño park, and N-120 road. Black bars represent the multi-model deviation.

**Table 1:** Main characteristics of the basin under scope (Miño-Sil basin to Ourense city), including the Miño river and its main tributary, the Sil river.

**Table 2:** Regional Climate Models (RCMs) driven by global climate models (GCMs) from EURO-CORDEX project. Perkins' test indicates those models with skills above 90% to reproduce measured values over the wet season (NDJFM). Twenty bins were considered in the Perkins' analysis. $P^{99}$ test indicates those models with a deviation below 25% when extreme values (above 99 percentile) were compared with measured data over the wet season. Models fulfilling both conditions are highlighted in bold.

**Table 3:** Historical (H) and Future (F) percentile values of river discharge ($m^3s^{-1}$) at Ourense city simulated for the RCMs marked in bold in Table 2.

**Table 4:** Number of days in a flood alert situation detected numerically for the Historical (1990-2019) and Future (2070-2099) periods in Ourense city. A day is considered under an alert situation when the river flow surpasses 2000 $m^3s^{-1}$ at some moment, following Fernández-Nóvoa et al. (2020). The percentage increase is calculated using the following equation: *Percentage_Increase=100\*(Future_Alert-Hisorical_Alert)/Historical_Alert.*

| Main characteristics of the basin under scope | |
|---|---|
| Location | *North-Western Spain* |
| Area (km$^2$) | *12822* |
| Mean slope (º) | *9.57* |
| Altitude range (m.a.s.l.) | *110-2100* |
| Mean altitude (m.a.s.l.) | *824* |
| Miño river length (km) | *134* |
| Mean slope of Miño river (m/m) | *0.0021* |
| Sil river length (km) | *234* |
| Mean slope of Sil river (m/m) | *0.0077* |

**Table 1:** Main characteristics of the basin under scope (Miño-Sil basin to Ourense city), including the Miño river and its main tributary, the Sil river.

| # | Global // Regional Climate Model | Perkins Test | P$^{99}$ Test |
|---|---|---|---|
| 1 | CNRM-CERFACS-CNRM-CM5_CLMcom// CCLM4-8-17 | – | – |
| 2 | CNRM-CERFACS-CNRM-CM5_RMIB-UGent//ALARO-0 | – | – |
| 3 | CNRM-CERFACS-CNRM-CM5_SMH// RCA4 | – | – |
| **4** | **ICHEC-EC-EARTH_CLMcom//ETH-COSMO-crCLIM-v1-1** | **Yes** | **Yes** |
| 5 | ICHEC-EC-EARTH_DMI//HIRHAM5 | – | – |
| **6** | **ICHEC-EC-EARTH_KNMI// RACMO22** | **Yes** | **Yes** |
| 7 | ICHEC-EC-EARTH_SMHI//RCA4 | – | – |
| 8 | IPSL-IPSL-CM5A-MR_DMI//HIRHAM5 | – | – |
| 9 | IPSL-IPSL-CM5A-MR_GERICS//REMO2015 | – | – |
| 10 | IPSL-IPSL-CM5A-MR_IPSL-INERIS// WRF331F | – | – |
| 11 | IPSL-IPSL-CM5A-MR_KNMI//RACMO22E | – | – |
| 12 | IPSL-IPSL-CM5A-MR_SMH// RCA4 | – | – |
| **13** | **MOHC-HadGEM2-ES_CLMcom// CCLM4-8-17** | **Yes** | **Yes** |
| **14** | **MOHC-HadGEM2-ES_CLMcom//ETH-COSMO-crCLIM-v1-1** | **Yes** | **Yes** |
| 15 | MOHC-HadGEM2-ES_CNRM//ALADIN63 | – | – |
| 16 | MOHC-HadGEM2-ES_ICTP//RegCM4-6 | – | – |
| 17 | MOHC-HadGEM2-ES_SMHI//RCA4 | – | – |
| 18 | MPI-M-MPI-ESM-LR_CLMcom// CCLM4-8-17 | – | – |
| 19 | MPI-M-MPI-ESM-LR_CLMcom//ETH-COSMO-crCLIM-v1-1 | Yes | – |
| 20 | MPI-M-MPI-ESM-LR_CNRM//ALADIN63 | – | – |
| 21 | MPI-M-MPI-ESM-LR_DMI///HIRHAM5 | – | – |
| 22 | MPI-M-MPI-ESM-LR_ICTP//RegCM4-6 | – | Yes |
| 23 | MPI-M-MPI-ESM-LR_KNMI//RACMO22E | – | – |
| 24 | MPI-M-MPI-ESM-LR_MPI-CSC//REMO2009 | – | – |
| 25 | MPI-M-MPI-ESM-LR_SMHI//RCA4 | – | – |
| **26** | **NCC-NorESM1-M_CLMcom//ETH-COSMO-crCLIM-v1-1** | **Yes** | **Yes** |
| 27 | NCC-NorESM1-M_CNRM//ALADIN63 | – | – |
| 28 | NCC-NorESM1-M_GERICS//REMO2015 | – | – |
| **29** | **NCC-NorESM1-M_IPSL//WRF381P** | **Yes** | **Yes** |
| 30 | NCC-NorESM1-M_KNMI//RACMO22E | – | – |
| 31 | NCC-NorESM1-M_SMHI//RCA4 | – | – |

**Table 2:** Regional Climate Models (RCMs) driven by global climate models (GCMs) from EURO-CORDEX project. Perkins' test indicates those models with skills above 90% to reproduce measured values over the wet season (NDJFM). Twenty bins were considered in the Perkins' analysis. P$^{99}$ test indicates those models with a deviation below 25% when extreme values (above 99 percentile) were compared with measured data over the wet season. Models fulfilling both conditions are highlighted in bold.

| Models | | 5 | 10 | 25 | 50 | 75 | 90 | 95 | 99 | 99.5 | 99.9 |
|---|---|---|---|---|---|---|---|---|---|---|---|
| | | | | | | **Percentiles** | | | | | |
| 4 | H | 42.4 | 50.2 | 89.2 | 223.5 | 388.2 | 525.4 | 661.2 | 1088.6 | 1304.6 | 2012.0 |
| | F | 39.6 | 48.8 | 79.1 | 204.8 | 375.9 | 516.7 | 644.3 | 1137.7 | 1449.1 | 2211.3 |
| 6 | H | 45.2 | 53.6 | 97.8 | 232.5 | 395.8 | 541.9 | 699.4 | 1128.6 | 1340.8 | 1992.0 |
| | F | 42.3 | 50.6 | 85.3 | 212.1 | 388.7 | 540.6 | 694.4 | 1255.5 | 1561.6 | 2211.7 |
| 13 | H | 40.4 | 49.1 | 77.5 | 213.4 | 387.2 | 519.5 | 658.7 | 1075.1 | 1347.1 | 2250.8 |
| | F | 37.9 | 47.0 | 70.0 | 197.3 | 349.9 | 470.5 | 604.9 | 1111.1 | 1481.3 | 3167.4 |
| 14 | H | 40.9 | 49.7 | 79.6 | 219.2 | 388.3 | 514.2 | 655.4 | 1132.0 | 1482.1 | 2673.9 |
| | F | 37.4 | 46.8 | 68.8 | 199.7 | 351.2 | 462.2 | 602.8 | 1096.9 | 1517.8 | 2815.6 |
| 26 | H | 41.9 | 50.5 | 86.2 | 233.1 | 406.5 | 579.5 | 725.8 | 1159.6 | 1379.4 | 1986.6 |
| | F | 39.6 | 47.7 | 75.2 | 211.5 | 373.0 | 516.5 | 632.4 | 1006.8 | 1314.9 | 2233.9 |
| 29 | H | 48.4 | 57.6 | 104.6 | 244.8 | 394.6 | 526.1 | 642.5 | 1006.4 | 1223.8 | 1625.2 |
| | F | 43.9 | 52.9 | 88.7 | 228.6 | 379.9 | 487.4 | 582.7 | 927.2 | 1069.3 | 1766.1 |

**Table 3:** Historical (H) and Future (F) percentile values of river discharge ($m^3s^{-1}$) at Ourense city simulated for the RCMs marked in bold in Table 2.


| Model | Alert Flood Situations | | |
|---|---|---|---|
| | **Historical Period (1990-2019)** | **Future Period (2070-2099)** | **Increase (%)** |
| **4** | 22 | *24* | *9* |
| **6** | 19 | *31* | *63* |
| **13** | 26 | *42* | *62* |
| **14** | 34 | *43* | *26* |
| **26** | 22 | *29* | *32* |
| **29** | 10 | *11* | *10* |
| **Mean** | 22±7 | *30±11* | *36* |

**Table 4:** Number of days in a flood alert situation detected numerically for the Historical (1990-2019) and Future (2070-2099)
periods in Ourense city. A day is considered under an alert situation when the river flow surpasses 2000 $m^3s^{-1}$ at some moment, following Fernández-Nóvoa et al. (2020). The percentage increase is calculated using the following equation: *Percentage_Increase=100*(Future_Alert-Hisorical_Alert)/Historical_Alert.*

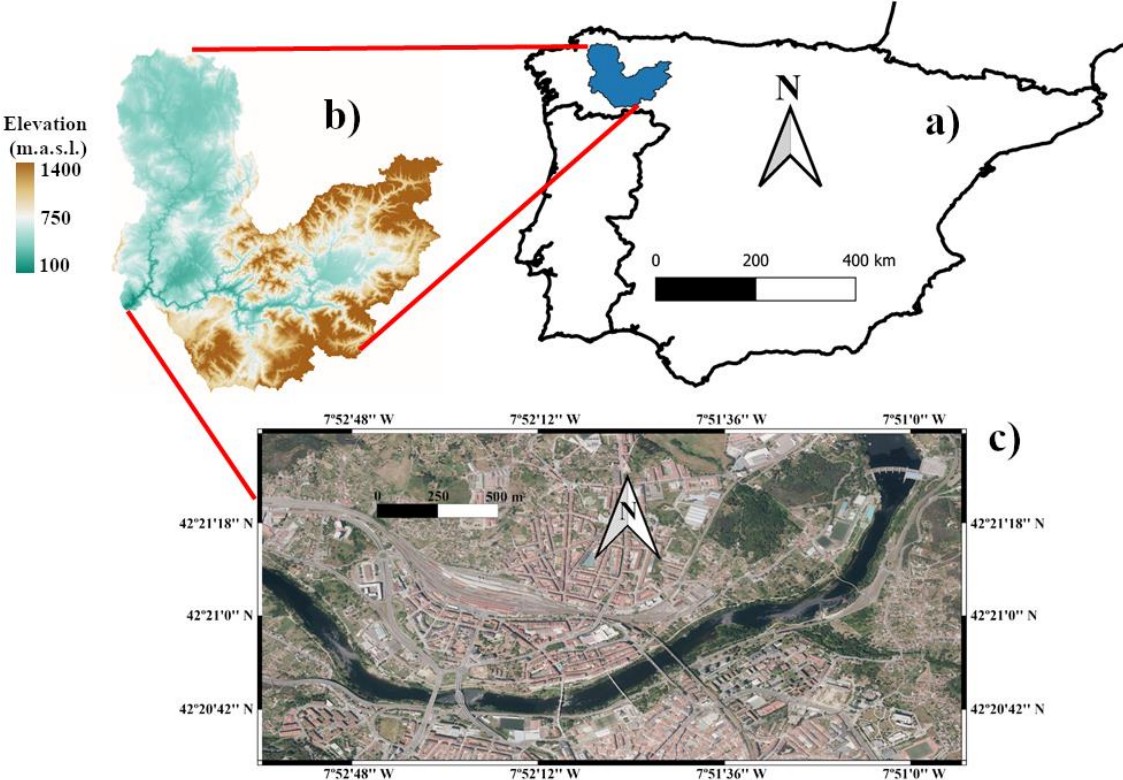


**Figure 1.** (a) Location of the Miño-Sil basin upstream of Ourense city. (b) Map of elevations of the catchment under the scope. The channel that flows from north to south represents the Miño river and the channel that flows from east to west its main tributary, the Sil river. (c) Area of study in Ourense city (PNOA courtesy of © Instituto Geográfico Nacional).


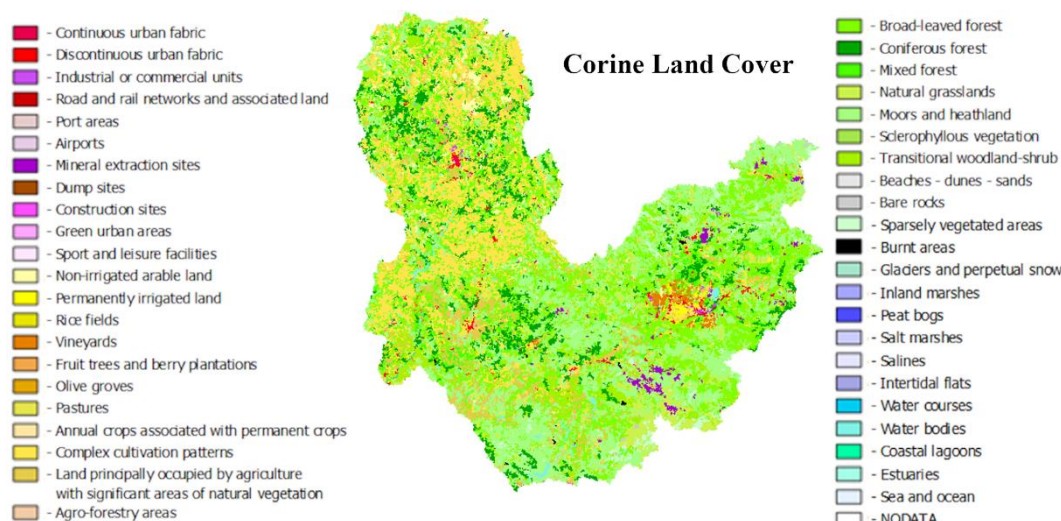

**Figure 2:** General land uses defined by Corine Land Cover and their distribution for the area under scope.


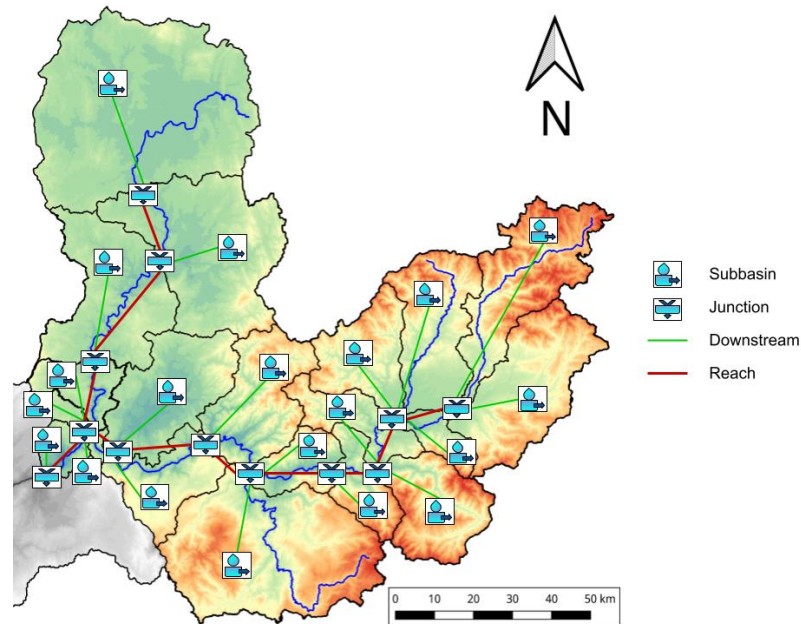

**Figure 3:** Map with the hydrological characterization of the basin under scope attending to its topographical features. *Subbasin* is referred to the divisions of the basin under scope, *Junction* is referred to the points in the flow network where multiple inflows combine, *Downstream* is referred to the point in the main river channel where each subbasin flows, and *Reach* represents the main river channels.

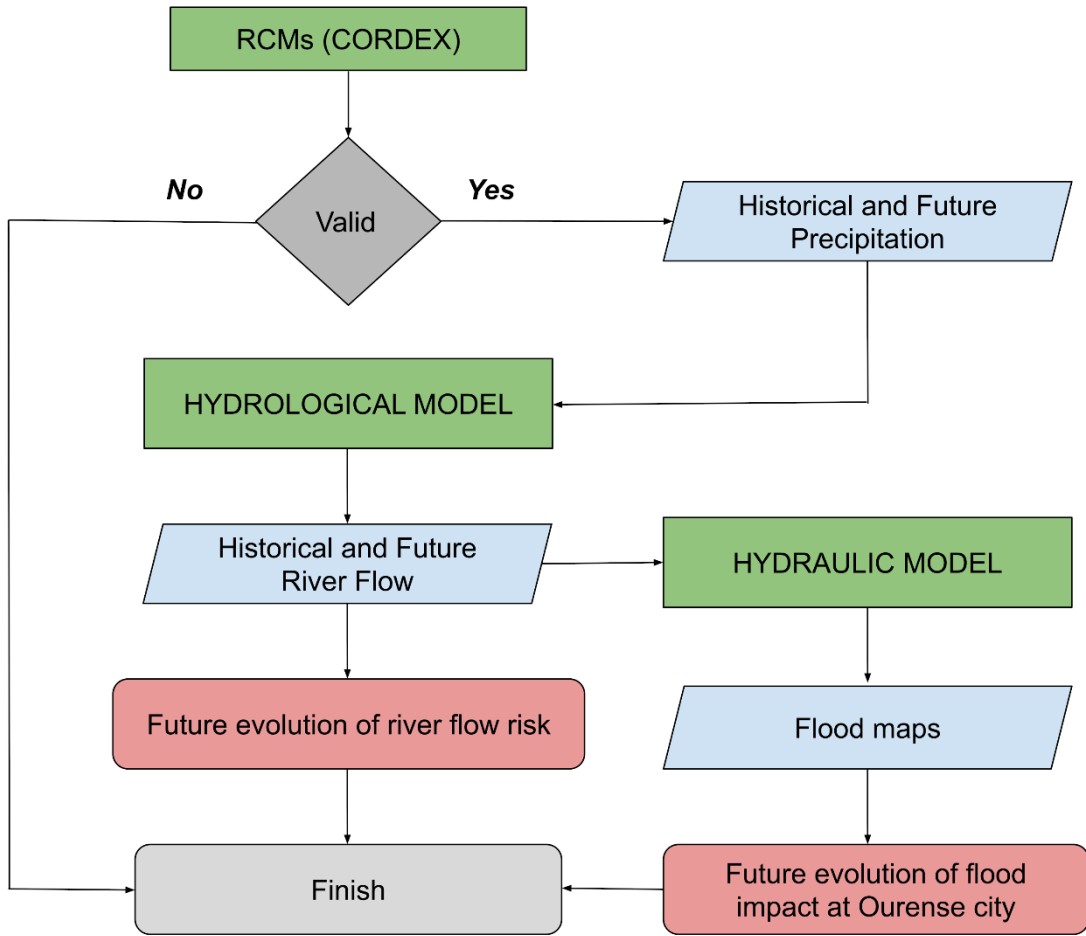

**Figure 4.** Flow chart of the developed methodology.


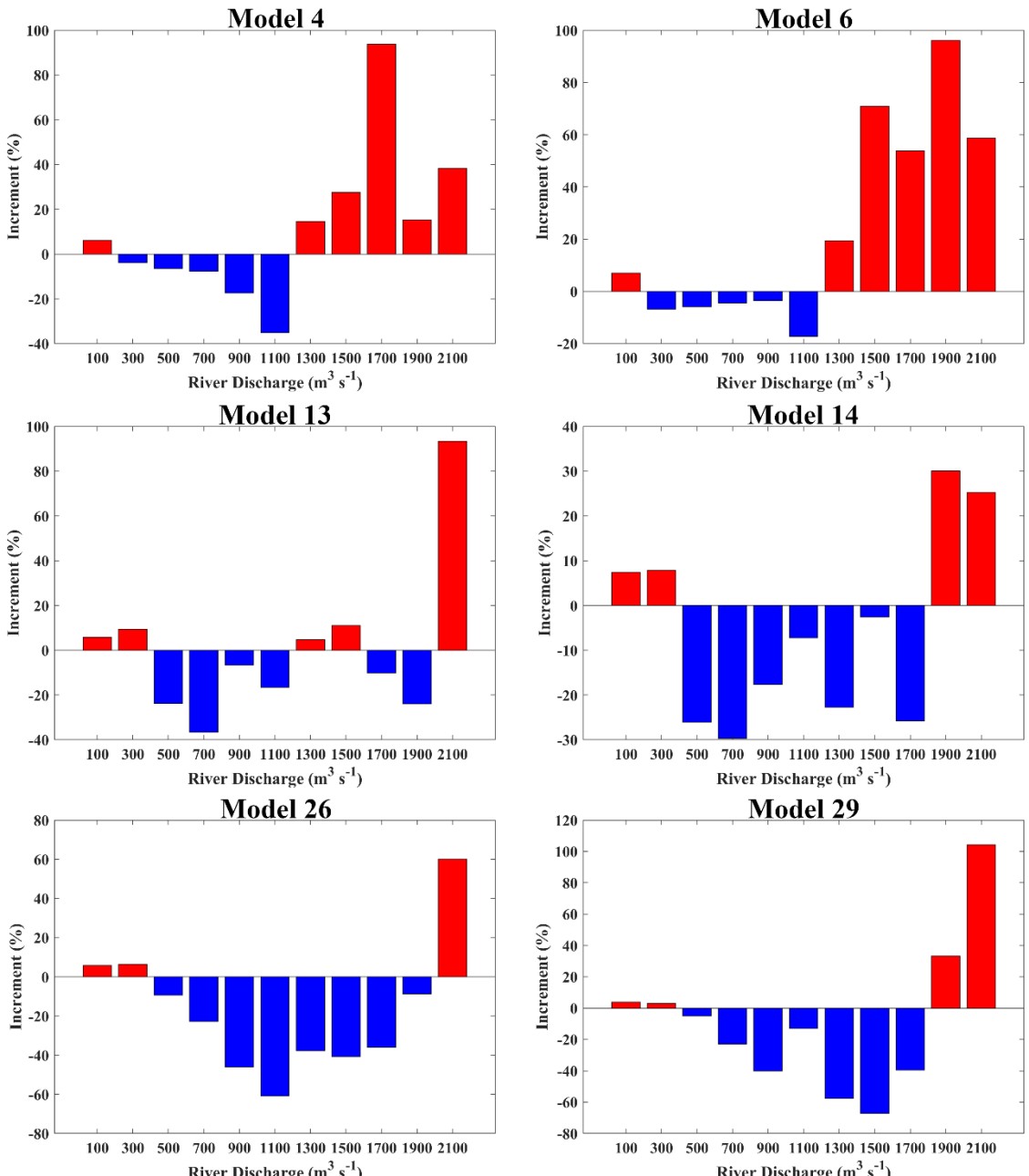

**Figure 5.** Percentage of change in the frequency of river flow ranges in the future with respect to the historical period for each valid model. The river flow data was distributed in ranges of 200 m³s⁻¹. Each value of the river discharge indicates the mean of the corresponding range (100 m³s⁻¹ corresponds to the range of 0-200 m³s⁻¹). In the last range (2100 m³s⁻¹), the river flows surpassing the alert threshold (2000 m³s⁻¹) are included. Positive (negative) values indicate increase (decrease) in the future.

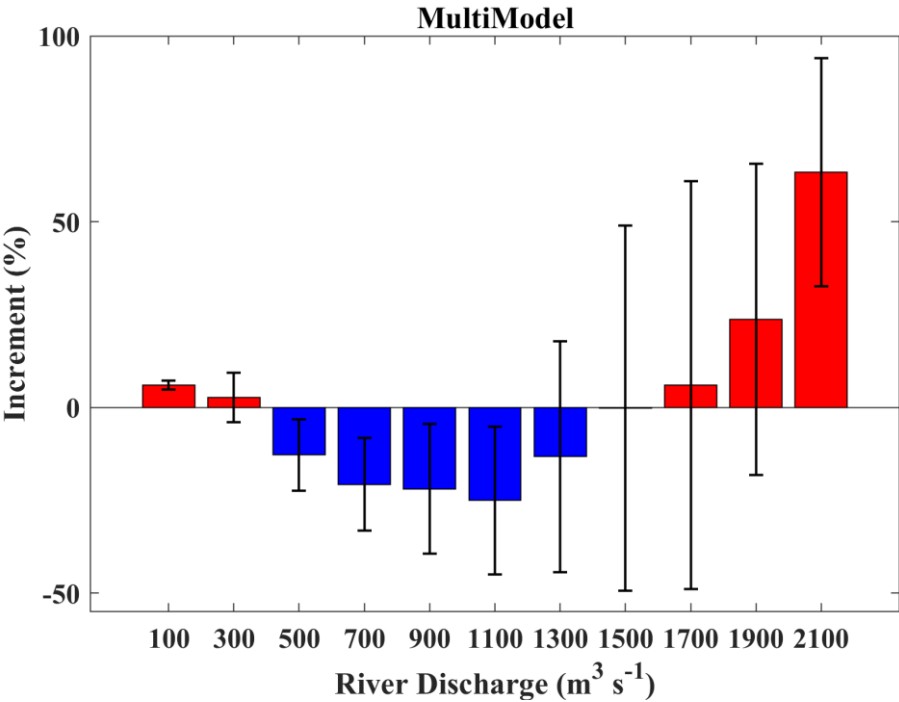

**Figure 6.** Percentage change in the frequency of river flow ranges in the future with respect to the historical period considering the average of valid models (multi-model). The river flow data was distributed in ranges of 200 $m^3s^{-1}$. Each value of the river discharge indicates the mean of the corresponding range (100 $m^3s^{-1}$ corresponds to the range of 0-200 $m^3s^{-1}$). In the last range (2100 $m^3s^{-1}$), the river flows surpassing the alert threshold (2000 $m^3s^{-1}$) are included. Positive (negative) values indicate increase (decrease) in the future. Black bars represent the multi-model deviation.


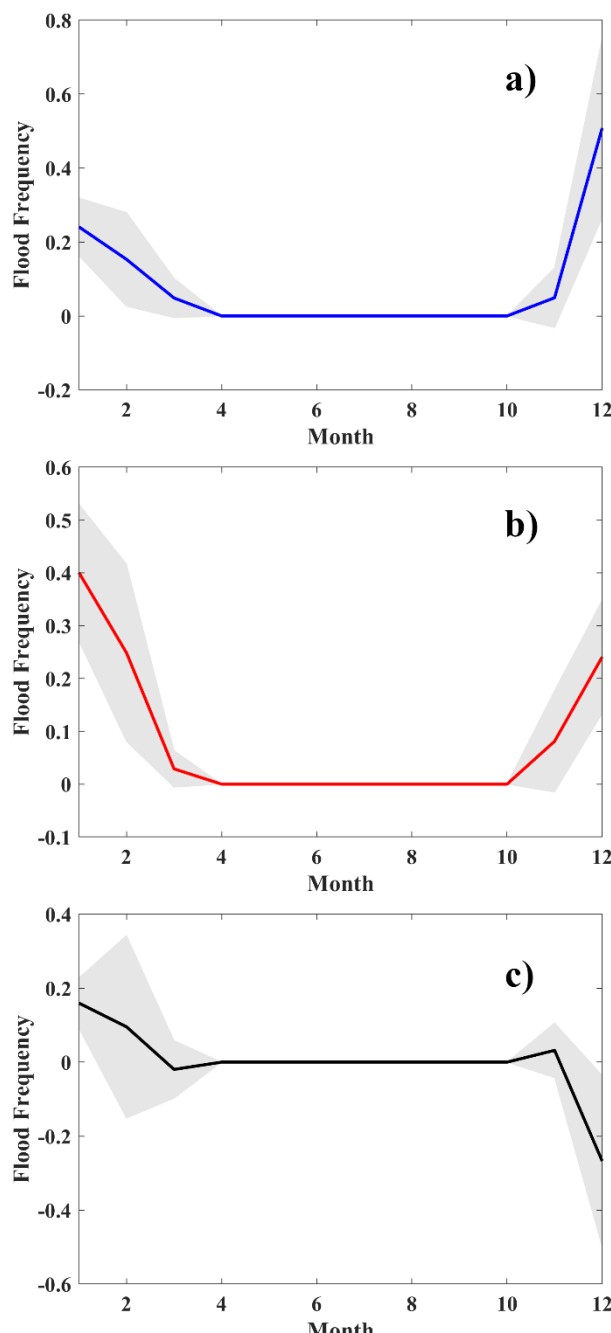

**Figure 7.** Multi-model mean of the monthly flood frequencies for the (a) historical and (b) future periods. Frequencies were calculated dividing the flood alerts detected in each month by the total number of alerts. Thus, the sum of the frequencies of all months is equal to 1. (c) Differences in flood frequency between historical and future periods. Positive (negative) values indicate an increasing (decreasing) of flood frequency in the future. Grey shadow areas represent multi-model deviation.


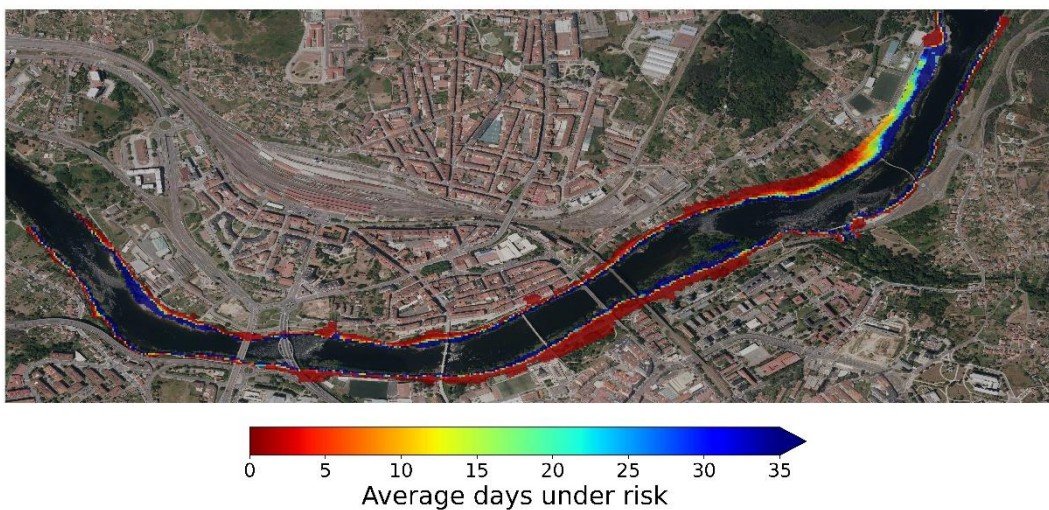

**Figure 8.** Multi-model mean of the number of days under hazard by floods for each pixel of the domain in the historical period. Only the area outside the riverbed was considered for the flood risk analysis. (Aerial picture courtesy of PNOA © Instituto Geográfico Nacional).

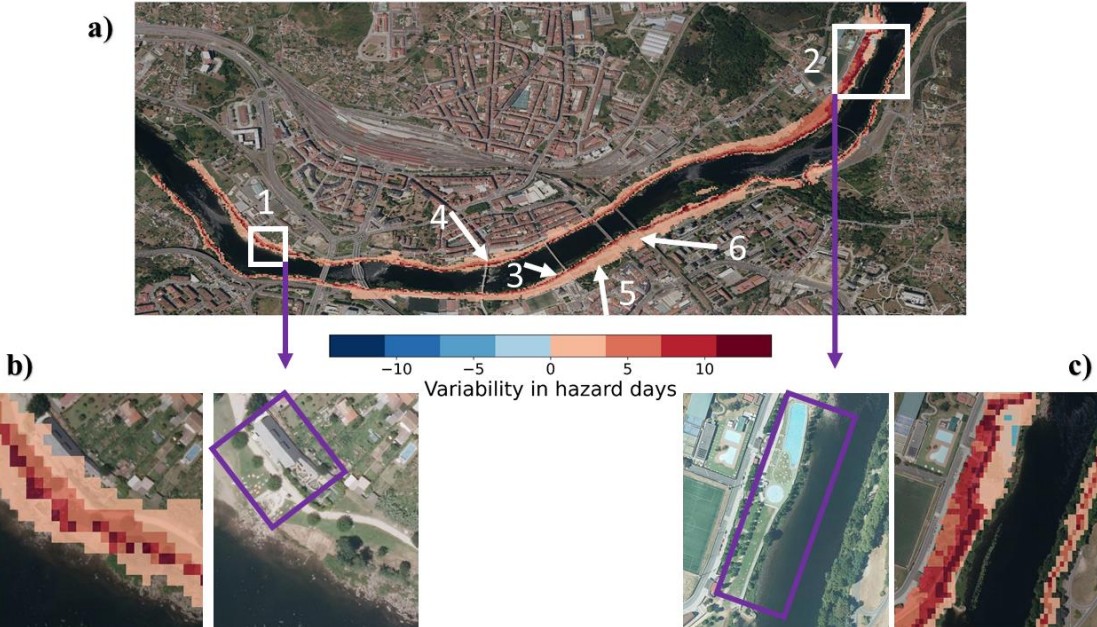

**Figure 9.** (a) Future variability in the number of days under hazard by floods in Ourense city. The increase (decrease) in the number of cases is obtained subtracting for each pixel of the domain the number of days under flood hazard situation in the historical period from the hazard cases in the future period. Thus, red (blue) colors represent the increase (decrease) in the flood hazard considering the average of all models. Only the area outside the riverbed was taken into account for the flood risk analysis. In lower panels two important areas of the city are shown in detail, (b) Chavasqueira thermal baths and (c) Oira park. (Aerial pictures courtesy of PNOA © Instituto Geográfico Nacional).

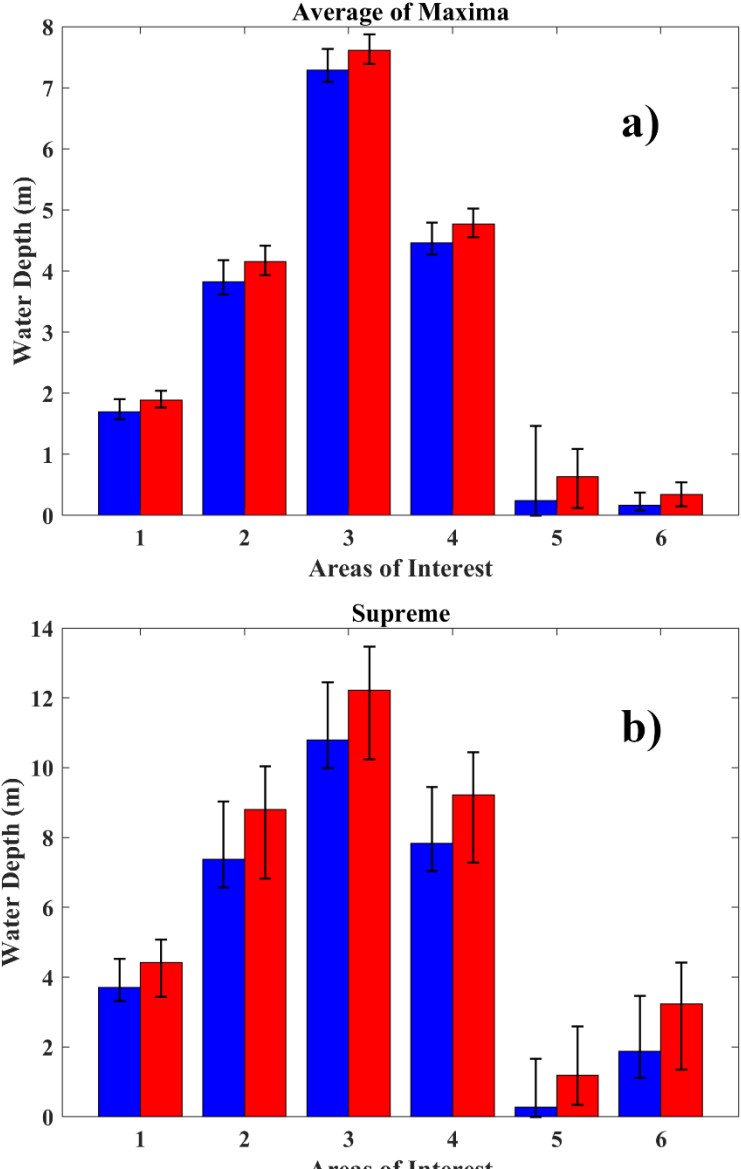

**Figure 10.** (a) Mean of maxima and (b) supremum water depths in important areas of Ourense for the historical (blue) and future (red) periods considering the average of all models. Areas under analysis are named from 1 to 6: Chavasqueira thermal baths, Oira Park, Antena fluvial beach, Ribeira de Canedo park, Miño park, and N-120 road. Black bars represent the multi-model deviation.