# Peer review of "Multiscale flood risk assessment under climate change: the case of"

_Natural Hazards and Earth System Sciences, 2022_

## Author Comment (AC1)

First of all, the authors want to thank the referees for the work and time devoted to review the manuscript. We know that all comments will serve to improve the quality and understanding of the work and we hope we have properly answered all the suggestions.

Lines are referred to the track-changes manuscript.

**Reviewer #1:**

The presented topic is quite interesting not only for local populations and policy-makers but also for the international research community. The authors presented a methodology that uses results from the CORDEX database to force a hydrological model that will feed a 2D hydraulic model. The methodology seems to be coherent and appropriated for the obtained results. The authors were careful in choosing the forcing conditions from the CORDEX project, comparing the historical results with observed precipitation data in the region. The numerical models selected are adequate, being well known tools fully developed and validated for different regions. The manuscript is well written, easy to follow and to understand, with up to date references. The authors seem to have previously experience with the topic, the region selected and the presented numerical models, which were previously configured in already published manuscripts.

However, there are some comments/issues that should be clarified.

The authors state in the introduction that "This provokes that the hydrological cycle regimes are mainly conditioned by the timing and position of winter storms, which in turn, are dependent on the NAO phase". However, is worthy to notice that there are previous studies demonstrating that the precipitation regimes in the Iberian Peninsula are not only dependent of a single atmospheric variability mode.

**This information was added and this part of the text was rewritten (lines 47-55).**

"The special synoptic conditions of the Iberian Peninsula, where the storm-tracks in the Northern Hemisphere can transport heat and moisture, are able to promote extreme weather conditions over this area (Peixoto and Oort, 1992; Trigo, 2006).  In particular, although the hydrological cycle regimes in the Iberian Peninsula are conditioned by several atmospheric variability modes (deCastro et al., 2006), the timing and position of winter storms are highly dependent on the North Atlantic Oscillation (NAO) phase, whose positive mode favors that the western Iberian Peninsula can be subjected to continuous intense large-scale precipitation during winter months, the main cause of the floods developed in the major rivers of this area (Lavers et al., 2011; Trigo et al., 2014)"

*The authors need to clarify the methodology section. Initially, the reader understands that the hydrological model will be forced which each one of the models that provide a good characterization of the precipitation for the study area, and the authors present Figure 3 and Table 2 with the individual results for each selected model. Then, the reader realized that an ensemble was constructed with the best CORDEX models and this ensemble will force the hydrological model. It is important to use an ensemble because, since the*

*hydrological model will be forced with results for another numerical model, the ensemble will avoid numerical inconsistencies and reduce the inaccuracies in the hydrological and hydraulic models. The early in the calculation that the errors are minimized, the smaller inaccuracies obtained. However, it will be necessary to include how the ensemble was constructed. It is a simple average or the authors considered a weighted mean? The weighted mean could provide more accurate results by considering the previous performance of the CORDEX numerical models in the weight.*

**The hydrological model was forced with the precipitation of each individual CORDEX model that surpassed the validation test. It should be noted that calculations were carried out for any particular model and the final results were averaged. Thus, the ensemble was constructed after hydrological-hydraulic simulations, for the results, by averaging the individual results provided by the hydrological (and hydraulic) model forced with each CORDEX model. In addition to the information provided by each model individually, the ensemble of the results allows offering a global view of changes minimizing the inaccuracies and errors as commented the reviewer. In climatological studies, the outputs of different models are never averaged prior to be used to force hydrological or hydraulic models. We have to take into account that several climate models do not reproduce the same meteorological situation at the same instant. Please, remember that weather models make predictions over specific areas and short timespans, while climate models analyze long timespans to predict how average conditions will change in a region over the coming decades.**

**As for the use of a weighted mean, it is difficult to establish the weights to carry out such a mean. The accuracy of the models was assessed in terms of two metrics and the "closely to reality" of the different methods was observed to depend on the metric, the area where the rain gage is located and the season.**

**This was clarified in the manuscript (lines 154-157 and 169-171).**

"Once the models that provide a good characterization of the precipitation for the study area have been selected, the precipitation provided by each one is used as an input in the hydrological model to simulate the river flow in the entire Miño-Sil basin. Thus, a hydrological simulation was carried out for each valid CORDEX model considering both historical (1990-2019) and future (2070-2099) periods".

"Some of the results will also be shown by averaging the individual results obtained after forcing the hydrological-hydraulic models with the data provided by each valid CORDEX model (multi-model), which minimizes the biases and uncertainties of each model (Pierce et al., 2009; Jacob et al., 2014)"

*The authors should better describe the procedures to construct the forcing with the CORDEX data. In the methodology it was not specify the version of the CORDEX project data. It is CMIP5 or CMIP6? Having a full range of numerical predictions, why the authors only selected the RCP8.5? If possible, it will be interesting to compare with a not so extreme scenario (RCP 4.5, for example).*

**The data from the CORDEX project used in this work were those corresponding to the CMIP5. This was clarified in the new version of the manuscript (lines 98-100).**

**Regarding the numerical predictions, we opted to analyze the most extreme scenario, the RCP8.5, in order to evaluate the most extreme changes and**

**implications that the climate change can cause in the study area in terms of floods. In this sense, a more conservative perspective towards the worst scenario is preferable for this type of applications and for the development of possible mitigation-adaptation measures. Although the comparison between different scenarios can also be interesting, authors consider that it is out of the scope of this study. Moreover, some of the CORDEX models validated for the study area have not available the RCP 4.5 prediction. In addition, in the report published by Schwalm et al. (2020) in Proceedings of the National Academy of Science, they found that, since the RCPs were developed, the historical evolution has been closest to that worst-case pathway. Thus, for the past 15 years, the greenhouse gas emissions have tracked most closely with those projected under RCP 8.5. To sum up, the worst case also seems to be the more realistic. Therefore, we decided to maintain the current analysis focused on RCP8.5 predictions. However, we acknowledge the reviewer for this recommendation because this is an interesting topic that may be addressed in future works.**

**This information was added in the new version of the manuscript (lines 96-104)**

"Historical (1990-2019) and Future (2070-2099) daily precipitation data for the area under scope were retrieved from the Regional Climate Models (RCMs) simulations carried out in the framework of the CORDEX project (http://www.euro-cordex.net/). EURO-CORDEX initiative offers simulations over the European continent considering global climate data from the Coupled Model Intercomparison Project Phase 5 (CMIP5) up to the year 2100, with more than 30 RCMs corresponding to the RCP8.5 greenhouse gas emission scenario. In this sense, some studies have detected that greenhouse gas emissions in recent years have tracked most closely with those projected under RCP 8.5, so this emission scenario seems to be highly realistic and a useful tool to assess future climate risks (Schwalm et al., 2020). EURO-CORDEX models provide daily data with sufficient spatial resolution (0.11°×0.11°) to adequately address the hydrological procedures of the area of scope (Garijo and Mediero, 2018; Lorenzo and Alvarez, 2020; Des et al., 2021)"

*It is not also clear the forecasting period. The authors referred that they used historical (1990-2019) and future (2070-2099) periods, and that the data has an hourly scale. However, it is not clear if the historical simulations and the future projections were done for an specific year or if the authors calculate an average for the full period. For historical conditions, using an average period to compare with the observed data is acceptable. However, a difference of 30 years in the projections could produce significant differences in the results.*

**All numerical simulations were performed for the historical period, considering the entire 1990-2019 period, and also for the future, considering the entire 2070-2099 period. Therefore, both periods analyzed have the same duration (30 years), in order to maintain the coherence for comparison purposes, as comment the reviewer. Thus, in both cases (historical and future), we run 30 years and compare the results for these periods. This was clarified in the manuscript. Only a shorter period was used to validate the CORDEX models due to the availability of measured precipitation data from pluviometers. In that case, the period 2008-2020 was used to validate CORDEX models with real data from pluviometers. However, in all the simulations, periods of 30 years (1990-2019 and 2070-2099) were always taken into account. This was clarified and specified in the new version of the manuscript (Lines 139-141, 156-157, 162-167).**

"The capability of each of the more than 30 RCMs models to represent precipitation over the area under the scope was tested comparing RCMs precipitation data and field data from pluviometers managed by MeteoGalicia for the common period 2008-2020".

"Thus, a hydrological simulation was carried out for each valid CORDEX model considering both historical (1990-2019) and future (2070-2099) periods."

"Changes in alert flood situations between historical and future periods were evaluated for each valid model. This analysis was carried out considering the total number of days under flood risk situation detected throughout the historical period (1990-2019) and the changes expected in the future period (2070-2099). The possible variation in the seasonality of the floods has also been evaluated. In addition, the river flow series obtained for the entire historical and future periods considered also allow the analysis of the expected changes in the general variability of river flow."

*Why the figure 5 presents the results for the whole year? The authors explained in the methodology that the period that they use to validate the precipitation data was for the wet season (November-March).*

**The November-March season was used to validate the ability of CORDEX models to reproduce precipitation patterns because it is when flood events occur in the area under study, and therefore, this is the period of most interest for the scope of this study. In fact, figure 5 corroborates this point; the flood events only occur during the November-March period. Once the models were validated, the complete years were simulated in the hydrologic procedure. Taking advantage of the information obtained, for sake of clarity we show the whole year in figure 5. This also allows corroborating when the flood season occurs and its evolution.**

*The authors mentioned that "The developed procedure takes between 2-3 weeks to execute each model". Please, include the characteristics of the computer used to run those models.*

**The simulations were executed on a computer with an AMD Ryzen 7 2700X processor, 32GB of RAM and a Nvidia RTX 3080 ti GPU. This information was included in the new version of the manuscript (lines 173-174).**

"with the simulations executed on a computer with an AMD Ryzen 7 2700X processor, 32 GB of RAM and a Nvidia RTX 3080 ti GPU"

*I recommend to the authors to include the limitations of the study. Is worthy to notice that there are several factors that could conditioning the future river flow that will reach a specific region, and not only the precipitation. The authors are representing the natural flow, but not changes in the man-made interactions with this flow. Changes in the aquifer capacity, in the river margins, in the soil characteristics, in the water use or in the hydroelectric production, among others, are non-easy predictable factors and will not be reproduced by the numerical models. However, they can have strong impacts in the floods.*

**We agree with the reviewer. The inclusion of the limitations of the study can help in the development of future studies. We add this information in lines 329-335.**

"The present work has some caveats, namely that the river flow can be also dependent of other interactions. In this sense, the present study tries to represent and analyze the evolution of the natural river flow based on the expected changes in precipitation patterns, as well as the implications in terms of the future floods. However, river flow is also influenced by man-made interactions, such as changes in river margins, soil characteristics, water use or the hydroelectric management, among others. These factors, which are non-easy predictable, can also have influence on the flood evolution. In this sense, the present study can suppose a basis to following works dealing with the analysis and inclusion, at some extent, of the influence of these other interactions in flood dynamics"

*Figure 1c should include the latitude and longitude*

**Done.**

[Figure]

*Figure 2: Future evolution of river flow risk instead of "risk river flow" and at Ourense city instead of in Ourense city.*

**Done.**

---

## Author Comment (AC2)

First of all, the authors want to thank the referees for the work and time devoted to review the manuscript. We know that all comments will serve to improve the quality and understanding of the work and we hope we have properly answered all the suggestions.

Lines are referred to the track-changes manuscript.

**Reviewer #2:**

The study addresses the analysis of the future evolution of river floods in the city of Ourense (NW Spain), where flooding of the Miño river can cause significant damage. In particular, the historical and future precipitation data from the CORDEX project are used as input in a hydrological model (HEC-HMS) which, in turn, feeds a 2D hydraulic model (Iber+). For each model, hydrological simulations were carried out considering both historical (1990-2019) and future (2070-2099) periods.

**Major comments follow.**

**In the Introduction the novelty of the study with respect to the state-of-the-art knowledge must be emphasized and the main objectives of the study must be better clarified.**

**We agree with the reviewer. The novelty of the study and the main objectives have been clarified in the new version of the manuscript (Lines 58-76).**

"For all the above mentioned, the main aim of this study is focused on the analysis of the future evolution of risk river flows in the regional domain delimited by the international Miño-Sil basin (NW Spain), and specifically, on the associated floods caused in the city of Ourense. Ourense is the local domain where flooding of the Miño river can cause significant damages (Fernández-Nóvoa et al., 2020). For that, the integration of hydrological-hydraulic models, together with specific information on flood thresholds in the area under scope, will allow a detailed analysis of those particular events capable of causing a significant impact in the study area. In this sense, most of the previous studies dealing with flood projections analyzed the expected changes in extreme flows and floods associated with different probabilities or return periods (Te Linde et al., 2011; Huang et al., 2013; Arnell and Gosling, 2016; Alfieri et al., 2017; Padulano et al., 2021). Here, the particular flows that effectively cause damage in Ourense city, not necessarily associated with characteristic return periods, as well as the specific areas that will be most affected by the associated floods, will be analyzed. This approach allows the analysis to be adapted to the particular characteristics of the area under scope, thus contributing to addressing one of the most important challenges facing the scientific community for the coming decades: assessing the implications of climate change on extreme events at the local level. To achieve these objectives, the river flows in the Minho-Sil basin will be simulated for an entire historical period of 30 years and also for an entire future period of 30 years using data provided by climate models, analyzing not only the changes in the probability of risk flow situations but also the evolution of flood risk in particular areas of the city. The results will provide detailed information to decision-makers to help them take accurate and effective measures to mitigate the damages associated with future extreme events. In addition, it is important to highlight that although the methodology developed is focused on the Miño-Sil river basin, it can be generalized to any basin and location, showing its potential in future applications."

Although in principle, the methodology seems appropriate, several details must be added to let the reader evaluate the correctness of the adopted approaches. In particular, the following key points should be better explained.

The capability of the EUROCORDEX RCMs models to represent precipitation over the area under investigation was tested by comparing RCMs precipitation data and field data by analyzing the entire distribution of precipitation data through the Perkins' test and also the extreme precipitation values through the P99 test. I assume that the Perkins' test is sensitive to the choice of the bin size and, in turn, the number of bins used to calculate the PDF. The authors should provide additional details on the test metrics and comment on this point, as well as on the advantage of this method with respect to statistical measures, such as bias, root mean square error, correlation, and trend analysis, commonly used to quantify model performance (see for instance doi.org/10.5194/nhess-20-3057-2020).

We agree with the reviewer that there are several analyses that can be suitable for validating the performance of climate models, depending on the scope of the study. Some works, such as the article recommended by the reviewer, use some statistical parameters such as bias, root mean square error, standard deviation, or mean values, among others, to validate climate models. However, these statistics are usually applied at monthly, seasonal or annual scales. In addition, some of them (e.g. means, standard deviations...) do not provide information of the entire data distribution, and therefore, a good fit in these statistics do not guarantee the adequate determination of some patterns of the data, which can have an important impact on the hydrological procedure. To develop our study, focused on flood analysis, we need to use the best available temporal scale, since floods are highly dependent on daily or even more precise time scales, and also corroborate the good determination of precipitation patterns, especially those referring to extreme events. Therefore, we need to validate those CORDEX models presenting a good skill to reproduce precipitation in terms of daily scale. In addition, as discussed in Perkins et al. (2007), the monthly, seasonal or annual analysis can hide biases or systematics errors that can be detected on the daily scale. Therefore, for the reasons commented above, we opted to maintain the validation of the CORDEX models using the PDFs, since, in addition, if the model is able to simulate an entire PDF, this also demonstrates the capability to deal with rare or very extreme values that can become more common in the future, as explain in Perkins et al. (2007). Therefore, we consider that this procedure is adequate for the purposes of our study. As for the number of bins, 20 bins were considered in the study (Table 2). Validation results show to be independent of the number of bins. We also complemented the test based on PDFs with the statistical analysis focused on analyzing the deviation of CORDEX models when representing extreme values, those able to cause flood situations. We consider that these analysis methods allow an adequate validation of the CORDEX models for the purposes of this study. Following reviewer recommendation, we provide additional details of the validation procedure, clarifying and explaining better this selection in the new version of the manuscript (Lines 139-153).

"The capability of each of the more than 30 RCMs models to represent precipitation over the area under the scope was tested comparing RCMs precipitation data and field data from pluviometers managed by MeteoGalicia for the common period 2008-2020. Although there are several valid approaches to validating CORDEX models (Perkins et al., 2007; Peres et al., 2020; Des et al., 2021) the present analysis is focused on testing both the entire distribution of precipitation data, the so called Perkins' test (Perkins et al., 2007) and also the extreme precipitation values (those above the 99 percentile, P99 test). One of the main

advantages of the Perkins' test, based on probability density functions and developed on a daily scale, is that the good fit between the measured and simulated data proves the capability of the model to detect extreme values that can be unusual in the historical period but more common in the future due to the implications of climate change (Perkins et al., 2007). This allows selecting the most appropriate models according to the scope of this study, since the extreme precipitation events are usually behind the most important river floods. In addition, the complementary analysis focused on evaluating the deviation of the models in relation to the extreme precipitation values reinforces the validation process. In this sense, only those models surpassing 90% in the Perkins' test and with a deviation less than 25 % in the extreme values were considered. In addition, in order to reinforce the detection of those models especially suitable for representing extreme events in the area under scope, the comparison was carried out over the wet season (November-March) when the flood events take place (Fernández-Nóvoa et al., 2020)."

The transformation of precipitation into the corresponding river flow was carried out using the semi-distributed model HEC-HMS. The authors should provide additional information on the hydrological model used for rainfall-runoff transformation (including the loss method for assessing the net precipitation). Also, please explain how the historical and future flows of the river were obtained on an hourly scale, given that precipitation data were at the daily scale.

**Additional details on the hydraulic modeling used for flood mapping are also required.**

Additional information related to hydrological and hydraulic models was provided in the new version of the manuscript (see section 2.3 Hydrological and Hydraulic Models).

**"2.3 Hydrological and Hydraulic Models**

The transformation of precipitation into the corresponding river flow was carried out using the semidistributed model HEC-HMS (Feldman, 2000; Scharffenberg, 2018; U.S. Army Corps of Engineers, 2018). This hydrological model was selected for being one of the most robust and widely adopted for hydrological procedures worldwide. In addition, this model has shown accurate results for the basin under scope as well as other nearby basins (Cea and Fraga, 2018; González-Cao et al., 2019; Fraga et al., 2020; Fernández-Nóvoa et al., 2020). In particular, Fernández-Nóvoa et al. (2020) shown an effective hydrological procedure for the area under scope using HEC-HMS and considering: i) the Soil Conservation Service (SCS) curve number for the rainfall infiltration calculations; ii) the SCS unit hydrograph to convert the rainfall excess in surface runoff; iii) the linear reservoir methodology to simulate the corresponding baseflow; iv) the Muskingum-Cunge method to deal with runoff propagation along the main channels. HEC-HMS is freely available on its official website (https://www.hec.usace.army.mil/software/hec-hms/).

The hydraulic model Iber+ was used to evaluate floods in Ourense city. Iber+ is a numerical tool that solves the 2D depth-averaged shallow water equations applying the finite volume method (García-Feal et al., 2018), and it is freely available on its official website (http://iberaula.es). In particular, the equations resolved by Iber+ model can be written as follows:

$$\frac{\partial h}{\partial t} + \frac{\partial h U_x}{\partial x} + \frac{\partial h U_y}{\partial y} = 0 \tag{1}$$

$$\frac{\partial hU_x}{\partial t} + \frac{\partial}{\partial x} \left( hU_x^2 + g\frac{h^2}{2} \right) + \frac{\partial}{\partial y} \left( hU_x U_y \right) = -gh\frac{\partial Z_b}{\partial x} - \frac{\tau_{b,x}}{\rho} + \frac{\partial}{\partial x} \left( v_t h\frac{\partial U_x}{\partial x} \right) + \frac{\partial}{\partial y} \left( v_t h\frac{\partial U_x}{\partial y} \right)$$
(2)

$$\frac{\partial hU_{y}}{\partial t} + \frac{\partial}{\partial y} \left( hU_{y}^{2} + g\frac{h^{2}}{2} \right) + \frac{\partial}{\partial x} \left( hU_{x}U_{y} \right) = -gh\frac{\partial Z_{b}}{\partial y} - \frac{\tau_{b,y}}{\rho} + \frac{\partial}{\partial x} \left( v_{t}h\frac{\partial U_{y}}{\partial x} \right) + \frac{\partial}{\partial y} \left( v_{t}h\frac{\partial U_{y}}{\partial y} \right)$$
(3)

where h is the water depth,  $U_x$  and  $U_y$  represent the averaged horizontal velocities, g is referred to the acceleration of the gravity,  $\rho$  is the density of the water,  $v_t$  is the turbulent viscosity,  $Z_b$  is the bed elevation, and  $\tau_b$  represents the bed friction. The bed friction is computed with the Manning formulation as:

$$\tau_{b,x} = \rho g h \frac{n^2 U_x |U|^2}{h^{4/3}} \quad \tau_{b,y} = \rho g h \frac{n^2 U_y |U|^2}{h^{4/3}} \tag{4}$$

Iber+ is an implementation in C++ and CUDA of the Iber model (Bladé and Cea, 2014). This new and optimized code achieves a two-order of magnitude speed-up by using graphical processing unit (GPU) and high-performance computing (HPC) techniques. These improvements allow to overcome the time constrained limitations of this type of climatological studies which require a large number of simulations. Iber+ has shown to provide accurate results in several studies conducted in the area under the scope and in nearby areas (González-Cao et al., 2019; Fraga et al., 2020; Fernández-Nóvoa et al., 2020).

The good performance of both models to resolve hydrological and hydraulic procedures for the study area allowed their integration for the development of an Early Warning System over the Miño-Sil basin with a good capability to reproduce real events (Fernández-Nóvoa et al., 2020). Therefore, attending to the results obtained in this previous study, where both models were calibrated and successfully validated for the study area, the same configuration of both models detailed in Fernández-Nóvoa et al. (2020) was maintained to develop the present study. The catchment schematization followed is presented in Figure 3."

**The process for obtaining the river flow on an hourly scale was clarified in lines 154-158.**

"Once the models that provide a good characterization of the precipitation for the study area have been selected, the precipitation provided by each one is used as an input in the hydrological model to simulate the river flow in the entire Miño-Sil basin. Thus, a hydrological simulation was carried out for each valid CORDEX model considering both historical (1990-2019) and future (2070-2099) periods. Although precipitation data from CORDEX is on a daily scale, it was added to the hydrological model on an hourly scale (dividing daily data by 24) in order to obtain river flows at this scale. Therefore, the historical and future flows of the river were obtained on an hourly scale"

**Minor comments**

**A table summarizing the physical features of the catchment (mean slope, altitude, river length, time of concentration) and a land cover map must be added.**

**Done (See new Table 1, Figure 2 and lines 82-88).**

"The area under scope is located in northwestern Iberian Peninsula (Figure 1a). It corresponds to the International Miño-Sil basin upstream Ourense city, encompassing more than 70% of the entire catchment, occupying near to 13,000 km2 (Figure 1b). The basin under analysis ranges in altitude approximately from 110 to 2100 m.a.s.l., with an average slope of around 9.57° (Table 1). The basin presents an important variability of land uses, although it is mainly characterized by moors and heathland (25 %), broad-leaved forest (23 %) and complex cultivation patterns (17 %), according to the criteria provided by CORINE land cover data (CLC, 2018) (Figure 2). Special attention is focused on the Ourense city, where Miño river floods can cause important damage (Figure 1c). The Miño river, which has an approximate length of 134 km to Ourense, presents a pluvial regime, with an annual hydrologic cycle characterized by minimum river flows during summer months and maximum flows at the end of autumn and winter, when the extreme flood events can occur (Fernández-Nóvoa et al., 2017; 2020)."

| Main characteristics of the basin under scope |                     |
|-----------------------------------------------|---------------------|
| Location                                      | North-Western Spain |
| Area (km²)                                    | 12822               |
| Mean slope (º)                                | 9.57                |
| Altitude range (m.a.s.l.)                     | 110-2100            |
| Mean altitude (m.a.s.l.)                      | 824                 |
| Miño river length (km)                        | 134                 |
| Mean slope of Miño river (m/m)                | 0.0021              |
| Sil river length (km)                         | 234                 |
| Mean slope of Sil river (m/m)                 | 0.0077              |

\_

**Table 1:** Main characteristics of the basin under scope (Miño-Sil basin to Ourense city), including the

 Miño river and its main tributary, the Sil river.

Figure 2: General land uses defined by Corine Land Cover and their distribution for the area under scope.

*Please provide additional details on the catchment schematization within HEC-HMS (i.e., number of sub-catchments, connections among them and so on).*

**Figure 3:** Map with the hydrological characterization of the basin under scope attending to its topographical features. *Subbasin* is referred to the divisions of the basin under scope, *Junction* is referred to the points in the flow network where multiple inflows combine, *Downstream* is referred to the point in the main river channel where each subbasin flows, and *Reach* represents the main river channels.

**Please clarify the meaning of "supreme water depths".**

To analyze the expected changes in maximum water depths reached in specific areas of the city subjected to floods, we determine the maximum water depth reached each day under flood conditions, and then, the average of these maximum values is calculated (mean of the maxima) and also the absolute maximum (highest value) is determined (referred as supremum value in the new version of the manuscript). Thus, the supremum value is referred to the highest water depth reached by water in each specific area taking into account all the days under floods. This was clarified in the text (lines 286-290).

"The multi-model mean of the maximum water depth reached during the floods in significant areas of Ourense was also calculated both for the historical period and for the future (Figure 10). For this analysis, the maximum value reached by water depth in each area for each day under flood conditions was determined, and the mean of these maxima (Figure 10a) and the supremum (Figure 10b) were calculated for each model and then averaged over all models."

L 70: replace "which supposes" with "encompassing".

Done.

Addendum to my previous comments It seems that the authors applied continuous hydrologic modeling with HEC-HMS. In this case, however, further input variables are needed, such as temperature. Please clarify!

The methodology applied do not need the input of more variables since is based on the Curve Number methodology, which requires only the precipitation data. This was clarified in the text (Section 2.3 Hydrological and Hydraulic Models).

---

## Author Response (AR2)

**First of all, authors would like to thank the referees for the valuable comments. Hereafter we hope to clarify the questions arisen**

Referee #2:

I would like to thank the authors for having considered my previous comments and suggestions in reviewing their manuscript. I only have one last criticism related to the fact that CORDEX daily precipitation data are simply divided by 24 to get hourly data to use as input in the hydrologic/hydraulic modeling. This distribution, besides being unrealistic, can strongly influence the results in terms of flood conditions. The authors should better argue that point and explain why this choice is better than any other hourly precipitation distribution in their study.

**We chose to apply this approach to the CORDEX data in order to use an hourly time step in the hydrological model and thus improve the accuracy of the results. In this sense, this allows obtaining an hourly river flow, which is more appropriate than considering only mean daily river flow. In addition, this also allows better performance of the hydraulic model. However, we agree with the reviewer that other precipitation distributions could have been used. As the reviewer comments, the precipitation distribution considered can influence the results, especially in those areas subjected to flash floods, as is the case of the Pyrenees river basins (Northeastern Iberian Peninsula), where the timing of the precipitation may be an important factor. However, the Miño river basin is hardly affected by flash flood processes, developing slower floods, so the fact of considering different precipitation distributions implies a lesser impact. In any case, it is important to bear in mind that the objective of this study is to compare historical floods from a climatological point of view with the future floods, in order to detect the future trends. In this sense, the different approaches considered to calculate past and future river flows and the associated floods, would not change the main results obtained within the aim of this work, that is, flood increasing in the future. Therefore, we opted to use the current approach, using thus the information provided by CORDEX without applying other artifacts, since the application of different precipitation distributions would involve other uncertainties and, as commented above, is less important attending to the scope of this work.**

**However, as authors consider these considerations to be important, a sentence clarifying these questions has been added in the manuscript (lines 155-158):**

*"Once the models that provide a good characterization of the precipitation for the study area have been selected, the precipitation provided by each one is used as an input in the hydrological model to simulate the river flow in the entire Miño-Sil basin. Thus, a hydrological simulation was carried out for each valid CORDEX model considering both*

*historical (1990-2019) and future (2070-2099) periods. Although precipitation data from CORDEX is on a daily scale, it was added to the hydrological model on an hourly scale (dividing daily data by 24) in order to obtain river flows at this scale.* *Although other precipitation distributions could be applied to obtain hourly resolution, we opted to use this approach in order to maintain CORDEX information without modifications taking into account the climatological perspective of this study, and also considering that the area under scope is affected by relatively slow floods where the distribution of precipitation has limited impact.* *Therefore, the historical and future flows of the river were obtained on an hourly scale"*